# An Eight-Week Zen Meditation and Music Programme for Mindfulness and Happiness: Qualitative Content Analysis

**DOI:** 10.3390/ijerph20237140

**Published:** 2023-12-04

**Authors:** Mi Hyang Hwang, Leslie Bunt, Catherine Warner

**Affiliations:** 1School of Health and Social Wellbeing, College of Health, Science and Society, University of the West of England, Bristol BS16 1DD, UK; 2Department of Buddhist Studies, Dongguk University, Seoul 04620, Republic of Korea

**Keywords:** Zen meditation, music, mental wellness, wellness programmes

## Abstract

Mental wellness can be related to healthier living, the learning process and working environments for people in the university community. A wide range of mental wellness programmes have been explored to provide students with pleasant and satisfying experiences. The purpose of this study is to explore the effects of integrating Zen meditation and music listening on the mindfulness and happiness levels of university music therapy students. A qualitative methodology was used, and data were collected through surveys and semi-structured interviews. To investigate various aspects of data regarding the role of the meditation and music (MM) programme for mindfulness and happiness, this study used thematic analysis within a qualitative research design. The findings of this study suggest that the 8-week Mindfulness Meditation (MM) programme is a potential approach for enhancing mindfulness, happiness and stress management. These results carry broader implications, particularly in terms of supporting mental health resources in higher education. Furthermore, the study contributes to the ongoing discussion regarding the positive impact of combining meditation and music to promote mental well-being. This integrated approach has the potential to strengthen coping strategies and further promote the integration of music and meditation practices in various contexts, including higher education.

## 1. Introduction

Meditation and music are each widely used in health, education and social community settings to promote general health and emotional and social well-being [1,2,3]. In recent years, there has been a growing interest in mental health among higher education [4,5,6]. In terms of academic life, mental wellness and psychological distress among university students are of increasing concern, especially since stress has become a part of their life due to various sources (e.g., exam anxiety, frustrations with achievements, deadline pressure, financial difficulties and emotional distress during the COVID-19 pandemic) [7,8]. Therefore, managing mental health and distress may largely affect students’ university life and academic achievements. There is an emerging evidence base suggesting that, under certain conditions at least, meditation and music, whether practised separately or in combination, can be effective in dealing with a variety of stress, anxiety-related conditions, emotional difficulties, and improving academic performance, as well as promoting a relaxed, centred and happy state of mind [9,10,11,12,13,14,15,16,17,18,19].

The feeling of happiness plays a vital role in the quality of life. Happiness is a subjective experience that encompasses positive emotions, life satisfaction, and overall well-being. When individuals experience a sense of happiness, it can have a profound impact on their physical health, mental well-being, and overall satisfaction with life [20]. Therefore, recognising and nurturing one’s happiness is crucial for leading a fulfilling and meaningful life. Various approaches have been explored to attain inner peace and happiness. Music can be a significant tool for enhancing well-being and regulating emotional happiness [21,22]. Music can serve as a source of joy and exert a significant influence on people’s perceptions of wellness and quality of life [23,24]. Similarly, meditation has been widely recognised as a beneficial intervention for promoting inner peace. It can help to cultivate abilities such as mindfulness, stillness and mental clarity [25,26]. These abilities have the potential to profoundly influence daily life experiences by improving self-control and emotional management [27].

This study aims to investigate how 8-week meditation and music (MM) programmes can contribute to cultivating a mindful state of mind and happiness, and examine the impact of these processes on students’ lives. There is a close relationship between mindfulness and happiness, as the practice of mindfulness has been shown to promote greater levels of happiness and well-being [28]. Mindfulness refers to the state of being fully present and accepting experiences without judgment, and it can contribute to happiness in several ways. First, being present in the moment allows individuals to recognise and appreciate positive emotions, fostering a greater sense of happiness in life. Second, mindfulness practices help individuals to develop greater emotional resilience and regulation. By observing their emotions with mindfulness, individuals can respond to challenging situations with equanimity, empathy and compassion, choosing more skilful and positive responses, leading to a greater sense of happiness. Third, a mindful mindset can reduce rumination, which refers to the unhelpful focus on negative thoughts and emotions. By practising mindfulness, individuals can break free from rumination and redirect their attention to the present moment, thus increasing their sense of happiness.

In this study, the feeling of happiness and mindfulness was explored using a combination of music and traditional Buddhist meditation based on Zen principles. The meditation practices incorporated elements such as present moment awareness, letting go, non-self, non-attachment and non-judgmental awareness, drawing inspiration from works like Nanamoli (1991) [29] and Hanh (2008) [3]. Additionally, the study also drew from the methods of Mark Williams and Kabat-Zinn’s mindfulness-based stress reduction (MBSR), specifically utilizing informal meditation techniques. These techniques, as outlined by Kabat-Zinn (2009) [30] and Williams (2008) [31], were employed to enhance psychological well-being and cultivate a state of happiness in the participants.

Indeed, numerous studies have demonstrated the benefits of meditation and music in managing various sources of stress and pressure in university life, as well as improving academic achievements and emotional stability [18,32,33]. Meditation and music interventions have been found to enhance present-centredness and emotional well-being [34]. Studies have highlighted how meditation and/or music can cultivate a positive mindset and mindfulness [35,36]. Regular meditation practice cultivates a state of present-moment awareness, which can improve emotional balance and cognitive abilities such as clarity of thinking. It allows individuals to find contentment and joy by fostering a greater connection to the present moment. Meditation can also promote a sense of hope and resilience by encouraging the release of negative thoughts and developing a deeper sense of a positive and optimistic perspective [37,38]. Music, on the other hand, has been shown to have a significant impact on happiness and pleasure. It can evoke feelings of inspiration and hope, serving as a source of comfort and encouragement. By uplifting mood and reducing stress, music can contribute to overall well-being. Moreover, when listened to attentively, music can foster a connection to the “here and now” experience and enhance a sense of mindfulness [39,40,41]. Therefore, music could contribute to overall well-being and positively impact various aspects of one’s life.

In this study, an 8-week MM programme was designed, integrating with meditation and music. In the past two decades, the benefits of combining meditation and music have been discussed, and mindfulness utilising music has been introduced in music therapy (MT) [42,43,44]. Certain similarities in purposes were found between meditation and music therapy, such as accessing expanded conscious states, achieving and strengthening the expanded ego, promoting the total development of personality and entering into a deeply relaxed state of consciousness [30,45,46,47,48]. Goldberg & Dimiceli-Mitran (2010) [48] discussed how meditation (e.g., maintaining a moment-by-moment awareness) enables people to observe how their inner mind works in the present moment during music listening.

Mindfulness-based music therapy programmes have been increasingly adapted with various groups, including musicians, university students and those with Huntington’s disease, drug and alcohol dependency and autism [49,50]. Lesiuk (2016) discussed the significant clinical benefits of mindfulness-based music therapy (MBMT), including reducing negative moods and fatigue, increasing focus and alleviating distressing symptoms among cancer patients. MBMT in cultivating self-exploration, self-development, self-understanding and self-transformation has also been discussed [49]. Van Dort (2015) further emphasised that combining principles of mindfulness (e.g., stillness, focusing on the present moment) and music can facilitate clients’ exploration of self-awareness and self-understanding [50]. Awareness and the state of mindfulness are indeed connected, as mindfulness practice enables individuals to become more aware of their thoughts and emotions. This increased awareness supports them in gaining insight into their mental and emotional patterns. By developing self-awareness, individuals can make conscious choices and exercise control over their lives, leading to a sense of autonomy and empowerment. This, in turn, contributes to a positive mindset and overall well-being.

Integrating meditation and music into mental health is not a new concept, and various aspects of meditation and music related to emotional wellness have been investigated [19,51,52]. As mentioned before, this study primarily focused on music-assisted Zen meditation as an important intervention. There are also various meditative approaches that may be suitable for integration with music. Therefore, exploring the integration of these two interventions can be a meaningful endeavour for those interested in mind–body treatment, alternative tools for mental and emotional health and overall well-being. However, despite several studies conducted in recent years to explore their combined approaches, the practice of integrating meditation and music is still relatively new and marked by uncertainty. The combination of music listening and meditation in a group-based format to explore mindfulness, happiness and quality of life of university music therapy students, particularly focusing on 8-week programmes, appears not yet to have been conducted. Additionally, there is a significant lack of discussion focused on how the combination of meditation and music impacts subjective happiness and mindfulness. This was the primary focus of this study.

Therefore, the purpose of the current study was twofold: First, to explore the state of mindfulness and the feeling of happiness cultivated through an 8-week structured programme. Second, to investigate how an 8-week programme (combining meditation and music) would affect psychological well-being, distress, self-control, positive states of mind, calmness, distraction and the overall experience of daily life for postgraduate music therapy students. Hence, the following research questions were explored:How does the supportive and structured 8-week meditation music (MM) programme contribute to increases in the participants’ feelings of happiness and mindfulness?How does the MM programme have a positive impact on participants’ ability to cope with emotional difficulties and ultimately improve their daily lives?

## 2. Method

### 2.1. Participants

The participants were invited to take part in this study to explore the effect of music and mindfulness on their ability to achieve mental wellness during the 8-week programme. The participants had to be 18 years of age and older and with no history of neurological or psychiatric illness, and people who were interested in meditation and music were recruited (See Table 1). The members of the group had diverse musical backgrounds and were familiar with therapy, as they were music therapy students. This MM programme experience was not therapy, but rather an exploration of meditation and music as a means of self-care. Participants were invited to participate via email invitations or face-to-face invitations. A small group size was preferred as an ideal size for a mindfulness-based practice, with approximately ten people in a group recommended [53]. Nine volunteers who were interested in the research project participated, three males and six females; and one dropped out due to personal circumstances; ages ranged from 24 to 59 years. Over 75% of the group did not have a particular religious practice.

### 2.2. Design

This study employed a qualitative research methodology utilising thematic analysis. Data were collected through semi-structured interviews, aiming to obtain an in-depth understanding of the role of music and meditation in promoting mental wellness. The study also sought to explore the relationship between the 8-week MM programme and the participants’ levels of mindfulness and happiness.

### 2.3. Setting and Procedure

The 8-week MM programme (See Table 2) took place in a quiet room with multiple chairs and tables. Ten yoga mats were laid on the floor, which was suitable for group relaxation. There was also an option to sit on a chair. Before starting the 8-week programme, a participant information sheet was given to groups of participants by email. Prior to the beginning of the session, all participants were asked to sign an informed consent form. The session took place at the end of a teaching day at the university. During the initial meeting, the participants introduced themselves and the session leader briefly explained the procedures of the 8-week MM programme. Participants were interested in meditation practice; however, as there were various kinds of meditation used in this MM programme, more detailed guidance was provided. This included a short introduction to meditation and, when necessary, practical examples were demonstrated, such as the ‘Breathing Hand’ and ‘Breathing Clock’. This programme emphasised the importance of both music and mindfulness and how they can be used to support each other. In each session, participants were given instruction on how to use mindfulness to enhance their music listening and how to use music to enhance their meditation practice.

The participants were informed that a session would consist of approximately 1–1.5 h. Each week, the programme consisted of: (a) a brief explanation about the session plan and setting intentions; (b) mini meditation; (c) music listening; (d) Zen meditation; (e) lying down meditation with music; and (f) sharing experience and closing (See Table 3 and Table 4). For self-practice, participants were given handouts each week relating to mindfulness meditation and music resources and were encouraged to monitor their home practice. Guided meditation files and music links, as used in the session, were sent via email for the students to practise at home once a day using their own streaming devices. During the 7th and 8th week, the individual interviews for each participant were conducted depending on the day the participants were scheduled.

### 2.4. Measurements and Data Collection

Interviews: Data were collected through audio-taped semi-structured interviews with each of the participants who attended the MM programme. The interviews consisted of both direct and indirect questions and ranged from 50 to 60 min in length. Semi-structured interviews were chosen because they would allow the participants a certain degree of freedom to explain their views and remain open to ideas [61]. The prepared interview questions were asked in an attempt to identify and clarify the effect of the MM programme for mental wellness. The interview questions were designed to identify and clarify the effect of the MM programme on mindfulness and happiness. Interview questions can originate from a variety of sources. The sources for this study include questions derived from various previous studies on meditation and music, as well as custom questions tailored to address the specific needs of using meditation and music (See Table 5).

### 2.5. Interventions

The main interventions of this study were mindfulness meditation, music listening and meditation with music (See the Table 2 above).

Music: The music was carefully selected depending on the session themes and types of activities. This is because different music brings different responses and results. Therefore, special attention is required to select music; for example, how the musical qualities (e.g., tempo, melody, amplitude, harmony, timbre) and duration of music can be matched with each meditation practice and activity. The majority of the music was relaxing music to provide a calming influence. However, other types of music were also used for some activities such as walking meditation. Every piece of music was short (between 4 and 9 min) in order for participants to not go too deep into a meditative state. Music genres included European classical music, new age music, binaural beat music, temple music and traditional Korean music. For the group listening, two wireless speakers were used. According to the theme of each week, participants experienced mindful music listening during each session and listened to background music before and after the session with selected music. Prior to the start of each session, participants were given handouts to read about all the resources of music in order to better understand the music resources.

Mindfulness Meditation: In this study, various postures of meditation were conducted, such as sitting, walking and lying down, and instructions were given before each session. Second, several key practices from the traditional Zen meditation subjects (Kammaṭṭhāna) were applied such as breathing meditation (Ānāpānasati), loving-kindness meditation (Mettā), compassion meditation (Karuṇā) and light meditation (Āloka kasiṇa) [58]. Third, Kabat Zinn’s MBSR techniques were also adapted (e.g., body scan meditation and resin meditation). Participants were instructed in Zen mindfulness meditation according to the guidelines presented, or guided by a recording file, especially before and after listening to music or while listening to music. Handouts were provided during each session relating to the practice of mindfulness and participants were encouraged to continue practicing at home. In addition, relaxation techniques such as progressive muscle relaxation (PMR) and yoga exercises were used at the beginning or final stage of each session. For instance, participants received a brief explanation of the rationale for PMR with a mindfulness state and how it could affect their body and mind. Then, they practised specific procedures following the links provided in the handout. Modifications were made to align this programme more consistently with the traditional mindfulness format and lasted for about 5–10 min.

### 2.6. Data Analysis

For the qualitative data analysis, thematic analysis (TA) was used as it is an effective and systematic approach to identifying important themes from a large collection of interview data. TA can provide a theoretical framework for the qualitative analysis of in-depth interview data [62]. In this study, Braun and Clarke’s six-phase approach to thematic analysis was employed: (a) familiarisation with the data; (b) coding; (c) searching for themes among codes; (d) reviewing themes; (e) defining and naming themes; and (f) producing the final report [62]. To analyse the interview data, hand-coding was performed. Potential codes and initial codes were identified, and each code was thematically collected. Then, potential codes and initial codes were identified, and each code was thematically collected. Then, potential broad themes and patterns were identified from the collected data. Finally, the main 6 themes were defined: (a) shifting the mindset; (b) calmness and relaxation; (c) focusing and centring; (d) feeling empowered and supported; (e) stress reduction; and (f) the role of music in meditative states.

## 3. Results

### 3.1. Qualitative Analysis

Through the semi-structured interviews, six main themes, along with their respective subthemes and example quotes, were identified (See Table 6 and Table 7).

### 3.2. Theme 1—Shifting the Mindset

Many of the participants reported experiencing a shift in mindset, such as moving from a restless and wandering mind to being more present in the moment, and from a judgmental mindset to one characterised by non-judgmental awareness.

“When I was listening, it’s kind of… I’m in the moment” (P2).

Mindfulness is inherently a practice of present moment awareness, and the ability to be fully present is one of the main benefits of mindfulness practice [30]. Participants appeared to recognise the advantages of embracing the ‘here and now experience’ (P5) and ‘being more present throughout the day’ (P1). Participants emphasised the significance of ‘being more present while doing’. For instance, P4 mentioned feeling more connected to their surroundings during walking meditation.

“I don’t know what the word is…but I felt a bit more in touch with what was going on around me… like noticing small things, noticing sounds…you are on your own…you notice them and then you let them go” (P4).

Participants frequently cited the advantages of cultivating a ‘positive mindset’, ‘practising non-attachment’ and embracing the concept of ‘letting go’. These practices provided a sense of peace and helped to calm overwhelming emotions and thoughts.

“When I drive back home, I feel more awake and more present and sort of revived a little bit… I feel refreshed when I drive back…I felt a lot more positive about the journey” (P2).

They discussed the positive impact of their shifting mindset on various aspects of their lives, such as their mindset, attitude and lifestyle. Many participants shared how the 8-week MM programme had become an integral part of their daily lives, and engaging in meditation and music practices played a role in cultivating a more positive mindset and attitude towards life.

“I noticed a positive change in my attitude after our session…” (P3).

Previous research has explored the relationship between mindfulness and attitude transformation. This encompasses the link between mindfulness and concepts such as letting go, fostering an accepting mindset, balancing active ‘doing’ with present ‘being’ modes, transforming negative thoughts and emotions into positive ones and shifting from restlessness to a state of calm awareness [63,64]. In this programme, there was no specific explanation of the theoretical or philosophical aspects of mindfulness. However, it appeared that the interviewees naturally grasped some fundamental principles of mindfulness through their own practice, including non-judgmental awareness, being present and letting go.

### 3.3. Theme 2—Calmness and Relaxation

Participants shared their experiences of feeling a sense of calmness and transitioning from restlessness to a more relaxed state while attending the MM programme.

“This was my first time trying mindfulness meditation through music, and it felt very relaxing” (P6).

“It made me feel calmer and more relaxed” (P2).

The connection between music and its psychological and physical stimulation of positive emotions and relaxation responses has been a frequent topic of discussion [13,65]. Participants reported experiencing increased calmness, which had a positive impact on their overall well-being and lifestyle choices.

“I know that it [music and meditation] is something that I should do on a daily basis and to be honest with this whole coronavirus thing going on what better time to start than now” (P4).

Interestingly, the link between ‘feeling relaxed’ and ‘preferring meditation’ was often mentioned. Individual preferences for meditation were varied. The most favoured and relaxed practices were lying with breathing practice, mini meditation and informal mindfulness meditation.

“I liked the compassion meditation and lying down meditation, they were really relaxing. I’ve never experienced anything like that before” (P3).

“They [raisins meditation and chocolate meditation] were really interesting…I’ve never ever done anything like that before…that was a nice surprise” (P5).

Several participants preferred the short version of meditation as they can quickly fall into meditation states and was not difficult to practise. However, some participants preferred longer meditation. P3 engaged in mindful observation of the back and spine areas, finding relaxation in the process. Both P1 and P4 expressed that ‘walking meditation’ held particular significance for them, and they expressed a desire to continue practising it, even on their way home.

“Lying down meditation, even if it was a longer one, I could really get the full benefit of it” (P3).

“I’ve never thought about really meditating when I’m walking and that’s really nice” (P1).

P3 had never practised meditation before but found the breathing exercises particularly helpful in coping with university life. The benefits of meditation, which can be easily integrated into daily activities, have been extensively explored [37]. Breathing meditation with music was consistently highlighted as a practice that helped participants experience a sense of calmness.

### 3.4. Theme 3—Focusing and Centring

The words ‘centredness’, ‘attention’ and ‘concentration’ were frequently mentioned by the participants. They found that music could help clarify and focus their thoughts, enhancing their ability to concentrate for longer periods.

“The music really helped me focus and concentrate better” (P1).

Some mentioned focusing on the music with mindful breathing techniques helped reduce ruminative thinking. Indeed, attention to the present moment is the main practice of mindfulness and by doing this, it can stop overthinking which causes tension or stress [19,30]. P1 felt that classical music in particular (e.g., Bach), helped in becoming more focused. The effect of classical music on attention and relaxation has been reported [66] and it has been used to be an effective resource such as measurements of attention [67].

“I think a lot and I can’t focus…but music helps me focus more on the here and now” (P5).

Participants expressed that the aesthetic experience of concentration played a primary role in their ability to experience mindfulness and cultivate a sense of calm in their lives. By immersing themselves in activities that required focused attention and concentration, such as engaging with aesthetic experiences like music or nature, they found themselves entering a state of mindfulness. This state of focused concentration allowed them to let go of distractions, connect deeply with the present moment and experience a heightened sense of calmness and tranquillity.

“When I listened to music during breathing practice, I felt like something was going [happening] deep inside of me…much more deeply” (P4).

This finding highlights the significance of aesthetic concentration in facilitating the experience of mindfulness and its associated benefits. A participant said, “It [meditation] made me listen to the music in a more attentive way” (P7). During the 8-week programme, concentration practice and meditation were integrated into each session, as well as assigned for home practice. This integration was based on the understanding that concentration and mindfulness are closely interconnected [68].

The literature extensively explores the state of mindfulness during meditation, further highlighting the relationship between concentration and the experience of mindfulness. Evans et al. (2017) discuss how meditation can strengthen the feeling of mindfulness and how mindfulness can be a useful tool for improving concentration levels [69]. In Zen meditation, practising concentration, insight and mindfulness are important vehicles for the purification of the mind [58]. The purified tranquillity and insightful mind could be related to the aesthetic experience. Johnston (1967) stated that the states of concentration achieved through meditation are noetic and ineffable and that noetic experiences can be achieved through mindful music listening [70]. Hence, it is assumed that music-assisted meditation can enhance the ability to focus on mindfulness and facilitate the attainment of noetic experiences, including a sense of centredness.

### 3.5. Theme 4—Feeling Empowered and Supported

Many interviewees expressed a desire to incorporate music and meditation into their lives because of the benefits these practices provide. The integrated method of meditation and music had identified effects, such as ‘empowering’ and ‘supported experiences.’ For instance, P3 expressed, “I feel strong”. Participants use music and mindfulness in various ways, depending on their situations and needs.

“It is really helpful and powerful” (P3).

“This experience is very healing energetically and emotionally and can really connect you to other people” (P1).

Some participants initially found meditation challenging, but with regular sessions, they became more comfortable with the practice. For example, P5 mentioned, “I feel I can access it and have a bit more direction in where I want to go with it”, while P4 said, “I’m starting to use mindfulness meditation, especially when I feel low in energy and distracted”. Consequently, it appears that participants perceived benefits from meditation and music, and they contemplated adapting these practices to enhance self-care and improve their quality of life.

Throughout the 8-week programme, participants received materials and information from the session leaders each week to facilitate their home practice. The intention was to motivate participants to continue their exploration of music and meditation, not only during the programme but also in their daily lives. These materials were delivered via email every week and distributed in person by the session leaders after each session. This approach created an opportunity for participants to practice more frequently, with some even incorporating meditation into their nightly routines before bedtime, when possible. While some participants turned to meditation to address insomnia, others found that frequent practice helped them to achieve a psychologically stable state and improved sleep. Several participants actively applied the principles from the 8-week programme to their everyday lives. During the interviews, it became evident that they were actively seeking various self-care methods.

### 3.6. Theme 5—Stress Reduction

The interviewees discussed personal challenges and how the MM programme assisted in coping with them. P3 said he encountered some challenging personal life experiences during the music and meditation sessions, but attending the sessions provided relief. P4 found that practicing mindful breathing and staying present with awareness can be very helpful in managing anxiety.

“I suffer from anxiety anyway. That’s why I wanted to try this [combination of music and meditation] … and having that weekly … it just gave me like a sense of calm for the rest of the week” (P6).

Five participants spoke of their sleeping problems and stated that these problems were the most stressful and difficult for them. P3 started practising meditation, both at university and at home, which included guided meditation for sleep to address night-time sleep issues, which were the most stressful times of the day for them. They commonly mentioned breathing meditation because they perceived that it improved their sleep quality. In fact, breathing-focused meditation has been frequently applied for insomnia [71]. A number of health benefits associated with breathing meditation have been reported such as improved sleeping quality and reduced mental and emotional stress. P6 felt that through mindfulness, there could be some restorative balance in body and mind.

“I wake up like three/four in the morning with so many ideas and I keep waking up” (P1).

“I meditated quietly while listening to music while lying down at night. This was the most relaxing time of the day for me” (P4).

P7 mentioned music performance anxiety and believed that using any form of mindfulness meditation during a performance might be worthwhile and meaningful. P3 said that mindfulness is a journey of self-discovery or self-awareness beyond stress management. Indeed, mindfulness practice increased self-awareness and through this, people often obtained a deeper sense of self-understanding and self-acceptance [64]. In addition, in terms of practical benefits, some used mindfulness during a long car journey because it made the journey easier: “This is the main impact of the programme for me” (P7). Mindful driving practice can reduce stress while driving [72,73]. Many studies provide evidence of the effects of mindfulness on specific attitudes and behaviours [74,75]. In this way, participants seemed to gradually realise that mindfulness may help to cope with their own stressors and personal difficulties.

“I feel better able to cope with stresses in my life” (P6).

“I would like to do a daily practice of at least 10 min of meditation” (P4).

Music and mindfulness have long been recognised, both separately and combined, as beneficial interventions for stress relief [76,77]. During the 8-week programme, many interviewees expressed a desire to incorporate music and meditation into their lives due to the benefits these practices offer. They sought to use music and mindfulness for stress management as these practices effectively helped them to gain control over their own stressful issues.

### 3.7. Theme 6—The Role of Music in Meditative States

One interesting finding in this study was that, for some participants, their favourite music proved to be a much quicker way to induce a meditative state compared to music they did not prefer. This suggests that the impact of music on meditation may be influenced by personal preferences. Participants had varying opinions on combining meditation with music, both with and without lyrics. Some participants (P1, P7) favoured music without lyrics, as lyrics can carry specific meanings that could potentially distract them from their meditative state. However, P6 expressed a different perspective, finding meditation songs with lyrics to be the best, as the lyrics, akin to poetry, enhanced their relaxation experience with the music. Additionally, participants mentioned the combination of meditation with oriental music. P1 expressed curiosity about temple meditation music and had a keen interest in exploring new cultural music. Similarly, P8 showed interest in Eastern temple music.

“It’s like opening a window to different ways of existing and living is a bit different from the likes of stressful society” (P8).

Some pieces of music were unfamiliar to the participants, yet they discovered that even unfamiliar music could enhance their meditation experience. As P4 put it, “All music I found fascinating, whether I liked it or not”. Classical music was generally favoured for its capacity to induce a state of calmness, with P1, in particular, feeling more attuned to Western classical music. However, one of the participants did not enjoy classical music due to their association with it during music exams, which reminded them of practising it for the exam. It is worth noting that music can evoke personal memories [78], and certain types of music may trigger vivid memories, like P2 in this case, who had a similar experience.

“I remember having a really good time. listening very deeply to the Chopin Nocturne” (P1).

Most participants in this study recognised that the MM programme included various types of music and meditation, and music preferences were not considered a significant issue for practising meditation. However, some participants felt that there was a relationship between the depth of meditation and their preferences. This suggests that personal preferences may play a role in the effectiveness of meditation practices involving music for some individuals. The relationship between personal music preferences and a deeper meditative experience is likely to have implications for the effectiveness and adoption of mindfulness practices involving music. Developing personalised mindfulness interventions that take individual preferences into account could help to optimise the effectiveness of mindfulness practices involving music and enhance their adoption among diverse populations. Meditative experiences and responses may vary depending on music preferences. However, in general, participants commonly perceived that most of the music in this programme helped them to reach meditative states.

“The music that you had was really helpful” (P1). “…more in-depth…thinking about the music” (P6).

Lastly, participants particularly experienced the emotional aspect of music during meditation. Musical experiences evoke emotional responses, and music influences emotions profoundly [40,79]. Participants mentioned that it was easier to delve more deeply into their emotions and connect with the emotional qualities within the music when they allowed space for meditative reflection. This may be attributed to their backgrounds as music therapy trainees, where it seems that music can lead them to a deeper meditative state. Similarly, a meditative state can enhance the depth of their emotional musical experiences.

## 4. Discussion

The study aimed to investigate the effects of an MM programme, which combined elements of music and meditation, on two main outcomes: mindfulness and students’ subjective mental happiness. The primary objectives were to understand how this programme impacted various aspects of participants’ well-being, including their ability to cope with emotional difficulties and daily life stress, as well as their overall mental wellness. The key findings of the study revealed that the MM programme had a notably positive impact on the psychological well-being of the participants. Specifically, the programme led to significant improvements in participants’ subjective happiness and mindfulness. Additionally, it was observed that the programme enhanced participants’ capacity to handle emotional challenges and effectively manage stress in their daily lives. Overall, these findings underscored the programme’s effectiveness in promoting well-being, mindfulness, and mental wellness among the students involved in the study.

During the 8-week programme, participants had various experiences of ‘state’ and ‘trait’ changes, and they discussed their experiences with the intervention and how it affected their daily lives. Participants reported feeling more mindful and peaceful during and after the MM sessions. However, it is also noteworthy that individuals had different experiences with aesthetic music and varied feelings of calmness depending on their preferences for types of interventions and the duration of practice. During the 8-week programme, participants actively engaged in discussions regarding their preferences and individual experiences. They openly expressed their feelings of mindfulness and shared their musical preferences, contributing to a more personalised and tailored approach to the programme.

This focus on individual experiences allowed participants to deepen their mindfulness practice and cultivate a greater sense of well-being. Additionally, they discussed feeling relaxed and expressed a preference for meditation as part of their practice. Interestingly, despite receiving positive external comments about their experiences, the participants demonstrated a thorough understanding and openness to the use of diverse types of music and meditation. They also expressed a willingness to explore culturally different music and engage in aesthetic meditation practices. This highlights their embrace of a wide range of experiences and their commitment to promoting cultural diversity and enrichment within the programme.

In relation to the duration of practice, it has been observed that meditation can elicit various levels of mindfulness states based on the depth of practice. Several studies have demonstrated that the length of mindfulness practice is primarily associated with an increase in the level of mindfulness [80]. Increases in the length and depth of mindfulness practice have been found to be associated with increases in the level of mindfulness [81]. These studies provide evidence that consistent and dedicated mindfulness practice can enhance one’s level of mindfulness, leading to a greater ability to cultivate present-moment awareness and engage in mindful behaviours. In this programme, each session comprised 4–5 activities for participants to access a variety of meditation and music resources, and a short version of mindfulness was frequently used due to limited session time.

Therefore, the outcomes may differ from previous studies that offered longer and deeper mindfulness practice [82], as the length of mindfulness practice significantly correlates with positive outcomes [83,84]. However, as seen in this study’s results, most participants felt the benefits of mindfulness in regulating emotions and reducing stress, and these findings are similar to previous studies [34]. Additionally, during the MM programme, various informal mindfulness meditations were used, such as mindful talking, mindful eating, mindful writing and mindful observation, which may enhance participants’ understanding of mindfulness and provide more opportunities to adapt mindfulness in everyday life.

As reviewed in the literature, happiness is considered one of the positive aspects of psychological functioning, and it is influenced by personal attitudes and behaviours [85]. In the context of the participants in this study, they reported experiencing changes in their attitudes and mood, such as letting go, acceptance, non-judging awareness and increased satisfaction with life. These experiences can have a beneficial impact on positive mood, emotions, feelings of happiness and overall mental wellness [86]. Furthermore, it is possible that the increase in happiness observed in the participants can be attributed to environmental factors. Being well-motivated students participating in a therapy training programme at a university of their choice, and engaging in group sessions facilitated by an experienced meditation teacher and practitioner, likely contributed to the positive outcomes. The supportive and conducive environment provided by the programme and the expertise of the facilitators can play a significant role in fostering personal growth, well-being and happiness.

What is notable in this study is that participants described various positive changes in their lives during the 8-week programme using terms such as ‘stress-free’, ‘positive’, ‘comfortable’, ‘letting go of painful feelings’, ‘moment-to-moment awareness’, ‘calm and clear-minded’, ‘relaxing’, ‘enhanced concentration’, ‘stability’, ‘grounding’ and ‘energetic’. These findings indicate that participants had a positive and stable mindset experience from the MM programme overall. Thus, the study objectives have been achieved by the findings: (1) The supportive and structured 8-week MM programme had a positive effect on increasing participants’ levels of mindfulness and happiness. (2) The combination of mindfulness and aesthetic music experiences had a positive impact on participants’ ability to cope with emotional difficulties and daily life stress, ultimately promoting their mental well-being and potentially leading to an improved academic performance.

In addition, there were discussions about reflections on the MM programme. First, many participants perceived that combining techniques effectively worked together: “The integration of music with mindfulness is such a good idea” (P7); If we didn’t have the music, I wouldn’t be able to get into that meditative state very easily because my mind is so busy” (P1). Second, in terms of group sessions, some participants found meditating with others to be particularly meaningful and mentioned it felt like a special connecting experience. P1 mentioned that continuous group meditation sessions helped in many ways, such as calming down and stopping excessive thinking. Third, the experience of weekly practice was discussed: “I like the structure of it (MM programme) and I think that it was particularly good” (P6); “It exceeded my expectations because of the positive outcomes I experienced” (P1). Some participants expressed their desire to continue practising through websites and online courses even after the programme ended: “Having the weekly session in my diary started to show me a few things which were important to me” (P5). Fourth, there were discussions about other factors that facilitated mindful music listening: “I think the way that the room and session are set up helps as well” (P4); “The yoga mats and circle of chairs provided a space to be still and quiet” (P5). Careful session preparation was necessary for this programme to ensure that it did not distract from stillness and deep listening, as mindful music listening requires a more focused way of listening.

Some topics and issues need to be reflected here which were emphasised by interviewees. First, participants described the mindful music listening experience as ‘focussing quite intently and deeply on the here and now’ and they said that this was similar to listening to music in music therapy. Particularly, they perceived deeper awareness along with focusing attention on features such as ‘aesthetic feeling’, ‘spiritual connections’, ‘stillness in my brain’, ‘aesthetic self’, ‘calming’, ‘become aware’, ‘feeling relaxed’ and ‘self-discovery’ while listening to music mindfully. Mindfulness and music therapy could share certain aspects, for example, ‘a sense of self-awareness in the moment during music listening’ [48] and being fully present and conscious of oneself while engaging in music listening [30,87]. Self-awareness is a clear awareness and ability to think about one’s own feelings, emotions, perceptions, attitudes and character [88,89]. Mindful self-awareness enables one to observe one’s own thoughts, behaviours and habits of mind (e.g., habitual worrying, habitual negative thinking) and it can help to cope with personal difficulty, stressors and everyday challenges by looking deeply and seeing clearly [60,90]. Self-awareness is intimately related to self-realisation. P3 was aware of a tense body during lying meditation and how it shifted from a stressed state to a more relaxed one. Releasing physical tension can help to relieve stress in the mind and body and P3 understood the benefits of relaxation on emotional and physical health. As such, mindfulness practice and/or music therapy enhances self-awareness and mindful observation and this may create a sustainable, healthy mind and personal wellbeing [91].

Second, the notions of satisfaction and motivation were emphasised. Participants had a strong intention to learn self-help techniques in order to cope with daily life stress [“I think too much and I have to stop doing that…I need any technique to clear my mind” (P2)]. Some remembered their protective behaviours and uncomfortable feeling about new things in the first week of the MM session: “When anything is new, we are a little bit on edge…but gradually I found that it was more and more helpful” (P7). Participants felt personal benefits and they were self-motivated in their own way. Those senses of satisfaction or reward may increase people’s autonomous reasons for home practice. In addition, self-motivation may improve the ability to think creatively [92]. “I am interested in combining meditation with music performance” (P5). As such, personal motivation and satisfaction seem to act to facilitate creative adaptation of music and mindfulness depending on their needs. Third, the use of mantra music or mantra meditation was discussed for MM resources as it can lead to deep states of relaxation. Two participants already used mantra meditation at home by themselves. A mantra (e.g., OM) was considered but it could be regarded as a particular religious practice or form of prayer; therefore, a mantra was not included in this programme in order to minimize any participants’ potential resistance. However, the benefits of mantra meditation and guided mantra chanting in group practice have been largely investigated [93], and the mantra can be a considerable resource for future work depending on the participants’ characteristics and session types.

In terms of instruments, several participants mentioned gongs, Tibetan singing bowls and crystal bowls after following the links provided in the handout. These instruments have unique sound vibrations and, from ancient times, healing aspects of sound have been reported [94,95]. Sound as a physical phenomenon can “create the optimal resonance between the vibrations of the singing bells and the vibrations of the specific receiver” [94]. Bidin et al., (2016) found that Tibetan singing bowls produced deep relaxation and reduced anxiety and this was clearly expressed by study participants, for example, “I felt the sound of the singing bowls joining and melting together and I sensed a breeze that wiped out anxiety and pain”; “I felt that I could again trust in my self-defence ability” [94] (p. 748). Gongs, Tibetan singing bowls and crystal bowls are traditionally used in sound therapy to induce depth relaxation and popular tools for meditation settings. Therefore, they may be useful instruments for creating unique mindfulness-based programmes.

Fourth, the advantage of group practice and group dynamics were discussed [“the group session is quite powerful” (P4)]. Several participants realised that there are differences between group and individual practice and group practice is more supportive to induce a deeper level of focus and relaxation. They spoke of feeling ‘more connected’, ‘collective energy in room’ and ‘group energy’. In fact, the value of the group format and a sense of group cohesion have been explored in other studies, such as “collective effort to make yourself feel better”, and “benefits of being on a shared journey” [14,96,97]. In this study, participants were all music therapy trainees, used to working in groups, and group dynamics may generate more easily in the form of mindful music listening. Several participants felt a depth of group energy experiences and this may possibly affect the positive changes in emotions and sense of well-being. Last, by the end of the interview process, the COVID-19 outbreak in the UK began and participants spoke about their feelings of fear, nervousness and uncertainty. Recently, a series of studies examined the impact of COVID-19 on emotions and mental health [98,99]. Participants anticipated that meditation and music could be useful tools for this unexpected difficult situation [“As we are in the middle of the pandemic, it feels important now more than ever to meditate and keep well, both in the body and mind” (P6)]. Several studies found the benefits of meditation practice (e.g., loving-kindness meditation, compassion meditation) and music (listening to music) in relation to the COVID-19 pandemic and these were considered as potential self-help interventions during times of crisis such as COVID-19 [100,101,102,103,104]. Participants frequently discussed the web-based resources and app for managing their stress (e.g., iTunes, Google Apps and Samsung health apps) [“I have a mindfulness App on my phone…it is really helpful” (P7)] [“It’s an app and it’s brilliant. You can there’s like, there’s big, long courses you can do over a few weeks there’s little short ones you can do” (P7)]. Recently, mental health apps are growing in popularity and the evidence for the positive effect of mindfulness and music apps has been reported in relation to self-acceptance, self-care, self-love and self-compassion [33,105,106].

Based on the findings presented, here are some suggestions for future directions and implications: First, further investigate the effectiveness of MM programmes in promoting happiness and emotional well-being in different populations, such as individuals with mental health disorders or those undergoing stress-inducing situations like academic or work-related stress. Second, explore the potential of incorporating different types of music and mindfulness practices into MM programmes, such as yoga or art therapy, to enhance the programme’s effectiveness in promoting mindfulness and well-being. Third, develop and implement MM programmes in educational institutions to help students cope with academic stress and promote their mental wellness. Fourth, consider the potential of MM programmes as a preventive measure for mental health disorders and incorporate them into public health policies aimed at promoting mental well-being. Fifth, evaluate the long-term effects of MM programmes on participants’ well-being and assess the sustainability of the programme’s positive impact.

Finally, it is crucial to compare the distinctions between the programme that combines music with mindfulness-based programmes and meditation programmes in isolation. In this study, many participants reported unique experiences during meditation with music. For instance, they found it easier to focus, achieve a deeper state of calmness and delve into more profound contemplative experiences, depending on the themes of the 1–8-week programmes. Indeed, these findings suggest that the integration of music and meditation plays a distinct role within the context of mindfulness and music-based programmes. This observation holds the potential to provide more substantial implications for researchers and clinicians in both the fields of music and meditation.

## 5. Conclusions

This study aimed to explore the effect of the MM programme on mindfulness and students’ subjective mental happiness. Overall, the study findings highlighted that the MM programme had a positive and meaningful impact on participants’ physical and psychological well-being. The programme also enhanced participants’ ability to cope with emotional difficulties and daily life stress. Support for student mental health and providing appropriate mental wellness services are increasingly recognised as new challenges facing higher education institutions [4,7,18,107,108]. This study could contribute to the development of a practical framework supporting mental health programmes, as well as expanding the range of potential interventions in the provision of healthcare areas with practitioners who have an interest in music and meditation.

Music and meditation interventions have each gained recognition for their benefits to health and wellness, and healthcare practitioners have carried out a great deal of work in this area [1,47,109,110,111,112,113,114]. There are various possibilities for modifying the MM programme for future research. For instance, comparing different settings such as individual or group sessions, conducting long-term follow-up research that includes pre- and post-treatment assessments, as well as follow-up quantitative approaches, exploring web-based courses for pandemic situations and examining integrative approaches that combine music, sound and mindfulness meditation. Control groups, including those using app-based mindfulness programmes or no-treatment controls, could also be included in future research to provide more robust evidence of the effectiveness of the MM programme. This research has the potential to contribute to the expansion of our understanding regarding the integration of theory and practice in the context of mindfulness meditation and music. It sheds light on how these practices can be effectively combined to enhance well-being and foster a sense of fulfilment in daily life. By exploring the benefits of the MM programme, this research highlights its potential to positively influence individuals’ daily lives by promoting well-being and empowering them to live more consciously and authentically.

## Figures and Tables

**Table 1 ijerph-20-07140-t001:** Demographic characteristics of study participants.

Characteristics	Demographic Groups	N (%)
Gender	Male	3 (33%)
	Female	6 (67%)
Age	20–39	6 (67%)
	40–59	3 (33%)
Religion	Christianity	2 (22%)
	No religion	7 (78%)

**Table 2 ijerph-20-07140-t002:** Interventions: 8-week meditation music (MM) programme.

Weeks	8 Week MM Programme			
1–10	Meditation	Music	Composer	Duration
Week 1	Week 1: “Recognising the Present Moment”			
	MM Introduction	Salut d’Amour Op.12	Elgar	3:13
	Being “In the Present Moment”	Music for Zen Meditation (Used Shorter Version: USV)	Tony Scott	44:33
	The Raisin Exercise	Meditation	Yuhki Kuramoto	3:26
	Breathing Relaxation	Korean Temple Music	Kim Young Dong	5:17
	The Body Scan	528 Hz Meditation Sound	Music healing	4:48
	Home Practice for Week One	5th Piano Concerto (2nd Movement)	Beethoven	7:40
		New Beginnings (USV)	Tim Janis	1:59:06
		Satori (USV)	Riley and Gabriel	15:48
Week 2	Week 2: “Mindfulness of the Breath and Body”			
	Postures for Meditation	Oboe Concerto in D, ‘Andante’	Strauss	9:26
	Mindfulness of Breathing	432 Hz Meditation Sound	Music healing	5:29
	Body Scan Meditation	3 Meditations (USV)	Riley Lee	23:32
	Mindfulness Exercise	Song for peace (1:40–2:44 except)	Kitaro	6:58
	Breathing Relaxation	Cello Concerto in C, ‘Adagio’	Haydn	9:56
	Home Practice for Week Two	Violin Concerto, Op. 77 2nd ‘Adagio’	Brahms	9:28
		Adagio and Allegro Op. 70	Schumann	8:51
		Five Variants on Dives and Lazarus (USV)	V. Williams	11:35
Week 3	Week 3: “Mindful Movement”			
	Relaxation Techniques	Deep Theta	Steven Halpern	6:49
	Mindfulness Exercise	Sonata No. 8 for piano (‘Pathetique’)	Beethoven	4:57
	3 Minute Breathing MeDi	Beyond The Time	Matsutani Suguru	5:54
	Mindful Movements	Simple Symphony (Sentimental Sarabande)	Britten	6:41
	Mindful Body Moving	Melody in F	Rubinstein	3:12
	Walking Meditation	The Swan	Saint-Saens	6:48
	Home Practice for Week Three	Concerto for two violins, II. Largo	Bach	6:48
		Adagio for strings	Barber, Samuel	9:27
		Bright future (USV)	Peder B. Helland	1:12:23
Week 4	Week 4: “Mindfulness of Sounds and Music”			
	5 Minute Breathing space	Piano Concerto No. 21 Andante	Mozart	5:51
	Simple Mindfulness	Mountain	Kim Young Dong	3:10
	Mindfulness of Music	Clarinet Concerto in A, 2nd movement	Mozart	5:32
	Mindfulness Sounds	Guided Mindfulness Meditation	Steven Halpern	5:32
	Lake Meditation	Lake Louise	Yuhki Kuramoto	3:26
	Home Practice for Week Four	Enchanted Lake, Op. 62	Anatoly Liadov	7:12
		Cinema Paradiso (00:00–3:32 only)	Haden and Metheny	8:01
		Piano Concerto No. 1, ‘Romance, Larghetto’	Chopin	12:29
Week 5	Week 5: “Exploring Difficulties Meditation/Turning Towards Difficulties”			
	Letting Go Meditation	Adagio in C BWV 564	Bach	3:59
	Breathing/Letting Go	Concerto for oboe in D minor	Marcello	4:41
	Self-Acceptance MeDi	Flying over the Clouds	Morricone	3:52
	Lying Meditation	Ave Maria	Bach/Gounod	8:00
	Home Practice for Week Five	Holberg Suite ‘Air’	Grieg	6:05
		Silk Road (6:17–12:10 only)	Kitaro	1:05:06
		From The End of The Evening Until Dawn	Cho Hye Lyung	2:56
Week 6	Week 6: “Befriending meditation”			
	Mindful Breathing	Adagio from Symphony No. 2 (USV)	Rachmaninov	15:09
	Self-Love MeDi	Romance for piano and violin, Op. 11 (USV)	Dvorak	12:46
	Loving Kindness MeDi	Canon in D (USV)	Kobialka	22:55
	Forgiveness Meditation	Meditation (Thais)	Massenet	5:14
	Mandala MeDi	Laudate Dominum	Mozart	4:25
	Home Practice for Week Six	Romance No. 2 for violin and orchestra	Beethoven	9:20
		Carnival of the Animals, ‘The Swan’	Saint-Saëns	3:06
		The Lark in the Clear Air (USV)	Kobialka	47:52
Week 7	Week 7: “Creative Mindfulness to Nourish Yourself”			
	Tradition meditation	Deep Theta Brainwave	Steven Halpern	5:28
	Mandala and Music	The Lark Ascending (USV)	Williams	14:44
	Drawing Your Breath	An Die Musik	Schubert	3:10
	Self Music and MeDi	e.g., The Notebook: Main Theme	Aaron Zigman	2:57
	Nourish Body and Mind	Mexican Memories (00:00–3:35 only)	Michael Jones	8:11
	Tangerine Meditation	Les Jours Tranquilles ‘The Quiet Days’	André Gagnon	5:50
	Grounding Techniques	Canon In D Major	Pachelbel	6:16
	Home Practice for Week Seven	Mandala Spin 10 Minutes (USV)	Charles	10:00
		Romance No. 2 in F, Moonlight Sonata (USV)	Beethoven	31:36
		An Anti-Frantic Alternative (USV)	Steven Halpern	45:27
Week 8	Week 8: “Music and Your Mind”			
	Mindfulness and Music	Oboe Concerto, 2nd movement	Albinoni	6:00
	Mindfulness and Breathing	Za-Zen (Meditation)	Tony Scott	2:05
	Create Your Own Music	Adagio in G Minor (USV)	Albinoni	12:05
	Mindful Creation	Innocence	Kenny G	3:58
	Breathing Space to Work	Forest Garden (USV)	Pachelbel	46:20
	Mindful Freedom	Mandala Meditation/Alpha Music (USV)	Charles	20:00
Plan(B)	Adaptation 1: “Living Mindfully/Mindfulness for Everyday Living (1)”			
	Cultivating Mindfulness	Melody in F	Rubinstein	3:12
	Three Steps Breathing	Arirang	Korean Song	4:15
	Mindful Eating	Appalachian Spring	Copland	3:09
	Mindful Writing	Cavatina	Myers	4:21
	Taking Care of Yourself	Liebestraum No. 3, Notturno	Liszt	4:39
	The Ways To Stay Mindful at Work (1)	Quiet Woods (USV)	Peder B. Helland	7:25

For all: The music and timings used for the listening part of the sessions were drawn from ‘Digital Compilations of Guided Imagery and Music (GIM) Programs’, compiled by Kenneth E. Bruscia for Barcelona Publishers. The researcher also drew on music from her own acquired resources [54].

**Table 3 ijerph-20-07140-t003:** Stages of the meditation music (MM) programme [sample].

	Stage	Example [Contents]	Duration	References	Key Terms
1	Introduction	Background music: Air on a G String by Johann Sebastian	5 min	Barger, 1979 [55] [rationale for music]	Bach’s Air on a G String, relaxation
2	Setting intentions	Mindfulness cards [adaptation: breathing meditation 4/4 breathing]	5–10 min	Arch & Craske, 2006 [56] [rationale for meditation]	Breathing practice on emotion regulation
3	Short version meditation or relaxation	Lazy 8 breathing in and out, meditation-assisted relaxation techniques [mini 3 min mindfulness]	5–10 min	Williams, 2018 [31] [rationale for mindfulness meditation]	Mindfulness, modes of mind
4	Mindful music listening	“Meditation” from the album Concertino by Yuhki Kuramoto	8–9 min	Dvorak & Hernandez-Ruiz [57] [rationale for music]	Attitude, mood, music characteristics
5	Zen meditation and mindfulness meditation	Traditional Buddhist meditation guided A-type meditation [1] (Kammaṭṭhāna)	10–15 min	Nanamoli, 1991 [58] [rationale for meditation]	Buddhist meditation technique
6	Lying down meditation with music	Traditional Buddhist meditation guided B-type meditation [2] (Kammaṭṭhāna) [music: Cello Concerto in C ‘Adagio’ by Haydn]	20 min	Nanamoli, 1991; Magill-Levreault, 1993 [29,58] [rationale for music and meditation]	Increased control, relaxation, meditation technique
7	Sharing	Personal experiences	10 min		
8	Closing	Closing and introduction of the next week session: mindful brain, mindful music listening, walking meditation, etc.	5–7 min	Goldberg, 2013; Hanh, 2008 [59,60] [rationale for music and meditation]	Music psychotherapy, classical music transforming, compassion
	Total		1–1.5 h		

**Table 4 ijerph-20-07140-t004:** The descriptions for each stage of the MM programme [sample].

Stage	Purpose	Key Activities
(1) Music as background	Creating a soothing atmosphere to facilitate participants’ entry into the programme	Playing background music to provide a sense of comfort and relaxation
(2) Simple meditation activities for setting intentions	Assisting participants in setting their intentions and preparing for the session	(a) Session plan and intention setting explanation: briefly explaining the session plan and helping participants set personal intentions or goals (b) Mini meditation: guiding participants through a brief meditation to focus their minds and find inner peace
(3) Meditation-assisted relaxation	Engaging in relaxation and meditation	Conducting relaxation and meditation exercises [with/without the influence of music]
(4) Conscious concentration with mindful music	Encouraging participants to cultivate conscious concentration while listening to mindful music	Practicing mindfulness and concentration while listening to specially selected music
(5) Traditional meditation for depth	Delving deeper into meditation through traditional techniques	Engaging in traditional meditation practices to explore inner tranquility
(6) Combining conscious concentration with music and meditation	Integrating conscious concentration with mindful music into traditional meditation practices	Simultaneously applying mindful music and traditional meditation techniques to enhance the meditation experience
(7) Sharing personal experiences	Providing participants with the opportunity to reflect on and share their personal experiences	Participants discuss and share their insights, emotions, and experiences related to the session’s music and meditation practices
(8) Closing the programme	Bringing the programme to a conclusion and expressing gratitude	Wrapping up the session, expressing gratitude to participants for their involvement, and providing any final remarks or instructions as needed

**Table 5 ijerph-20-07140-t005:** Interview questions.

	Sample of Interview Questions
1	What was your overall experience with meditation and music in this programme?
2	How do you think meditation and music can help to foster a positive mindset?
3	What were your thoughts about learning and practising mindfulness meditation?
4	What do you think of meditative and mindful music listening related relaxation responses?
5	What do you think of the personal effect of mindfulness meditation in daily life?
6	What do you think of integrating the practice into your daily life?
7	How does the MM programme affect your daily life ?
8	What was your experience in this 8-week MM programme?

**Table 6 ijerph-20-07140-t006:** Operational definitions of categories for participants statements.

Themes	Definition
Shifting the mindset	Statements related to a change in thought patterns, perspectives or mental states
Calmness and relaxing	Statements indicating a sense of tranquility, peacefulness or relaxation
Focusing and centring	Statements related to enhanced concentration, mental clarity and inner balance
Feeling empowered and supported	Statements reflecting a sense of confidence, self-empowerment and emotional support
Stress reduction	Statements mentioning the reduction or alleviation of stress, tension or anxiety
The role of music inmeditative states	Statements discussing how music contributes to achieving meditative or mindful states

**Table 7 ijerph-20-07140-t007:** Themes and subthemes from qualitative data and example quotes.

Themes	Example Quotes
Shifting the mindset	“Meditation really helped me because I’m not very good at switching off … my mind always racing” (P1) “I noticed a positive change in my attitude after our session…” (P3) “I was very anxious I had problems with anxiety…Breathing meditation was very helpful in getting me out of those kinds of situations. Nowadays, I feel a lot better” (P4)
Calmness and relaxation	“I feel a lot more relaxed it made me notice how tired I was through doing the relaxations” (P3) “While listening to music, I was able to feel a sense of calmness in my mind” (P1) “I noticed in my body…I noticed tension and then I’m able to let go and relax” (P5)
Focusing and centring	“I was able to focus more on my inner emotions” (P1) “Music really helped me focus and better concentration” (P4) “I found the music comforting, and it became a really useful thing to focus on when breathing in and out and relaxing into a more meditative state” (P3)
Feeling empowered and supported	“I would say the emotional aspect of the music come through much, much stronger, stronger” (P6) “I found them very healing and almost like a relief from the day” (P8) “It was, for me really useful to have space and time like dedicated to actually practising it” (P7) “I was taking sleeping tablets … I started doing a lot of those breath work before I went to bed and found that doing some really long, deep breathing at night. That really helped me” (P5)
Stress reduction	“I really always stress about my two-hour journey home and having that lie down beforehand. Yes, absolutely. It helps so much” (P5) “Sometimes I get too stressed when I breathe or during the lectures as well sometimes it’s too much information so just try to avoid any sound going around me and just breathe in and out” (P3)
The role of music in meditative states	“I like the music. Especially today. I think that was the best music… makes me more mindful and I can easily focus on my body” (P1) “When I listened to Eastern music, I felt a more peaceful state of mind. It was like the feeling of going on a journey to a new place” (P6) “The (classical) music was great just to help, with the calming down” (P7)

## Data Availability

Data are contained within the article.

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
