# Peer review of "An Eight-Week Zen Meditation and Music Programme for Mindfulness and Happiness: Qualitative Content Analysis"

_ijerph, 2023, doi:10.3390/ijerph20237140_

Round 1
Reviewer 1 Report (Previous Reviewer 2)
Comments and Suggestions for Authors
Dear Sir/Madam,
Thank you for inviting me to review the manuscript entitled "8-Week Zen Meditation (禪) and Music Programme for Mindfulness and Happiness: Qualitative Content Analysis" (manuscript ID: ijerph-2592642) submitted to the International Journal of Environmental Research and Public Health.
The study examines the effects of integrating Zen meditation and music listening on the mindfulness and happiness levels of university music therapy students. The study represents a valuable study in terms of conceptualization, planning and findings. My observations are presented below:
(1) The study title, generally, should not start with a digit. I suggest changing it ‘an eight-week’.
(2) The abstract is sufficient. There is contradiction in method written in the title and mentioned in the abstract and the methods section. I suggest adding some interpretations and implications of study findings.
(3) The introduction is good. There is no need to repeat the study objectives. Please the numbered objectives.
(4) The methods section also needs minor changes. Please the sources from the interview questions have been prepared and how the pre and post effects of intervention were assessed. The data analysis is not clearly described.
(5) The results section is ok.
(6) The discussion section needs minor improvements. As a rule, the discussion should begin with the description of the study objectives and subsequent key findings. They are then interpreted in the light of existing theoretical and empirical knowledge. Novel findings should be highlighted. The interpretation of the results needs to be improved. The implications of the study results need to be highlighted.
(7) The references are ok.
In conclusion, the study is a good work and its results can contribute to the field of knowledge. However, small changes are recommended. In its current form, the manuscript can be accepted after minor changes as suggested above.
I thank the editor again for the invitation.
With best regards,
Reviewer
Author Response
Dear Academic Editor,
Thank you for your valuable feedback on our manuscript. We have carefully considered your comments and have made the necessary revisions to address them. Here are our responses to each of your points:
Reviewer (1)
Q1. The study title, generally, should not start with a digit. I suggest changing it ‘an eight-week’.
Amended 1. An Eight-Week Zen Meditation(禪) and Music Programme for Mindfulness and Happiness: Qualitative Content Analysis
Q2. The abstract is sufficient. There is contradiction in method written in the title and mentioned in the abstract and the methods section. I suggest adding some interpretations and implications of study findings.
Amended 2. (Adding part) The findings of this study suggest that the 8-week Mindfulness Meditation (MM) programme is a potential approach for enhancing mindfulness, happiness, and stress management. These results carry broader implications, particularly in terms of supporting mental health resources in higher education. Furthermore, the study contributes to the ongoing discussion regarding the positive impact of combining meditation and music to promote mental well-being. This integrated approach has the potential to strengthen coping strategies and further promote the integration of music and meditation practices in various contexts, including higher education.
Q3. The methods section also needs minor changes. Please the sources from the interview questions have been prepared and The data analysis is not clearly described.
Amended 3: (1) Interview questions can originate from a variety of sources. The sources for this study include questions derived from various previous studies on meditation and music, as well as custom questions tailored to address the specific needs of using meditation and music.
(2) In this study, the pre- and post-effects of the intervention were not discussed
(3) Potential codes and initial codes were identified, and each code was thematically collected. Then, potential broad themes and patterns were identified from the collected data. Finally, the main 6 themes were defined: a) Shifting the Mindset; b) Calmness & Relaxation; c) Focusing & Centering; d) Feeling empowered and supported; e) Stress Reduction; and f) The role of music in meditative states.
Q4: The discussion section needs minor improvements. As a rule, the discussion should begin with the description of the study objectives and subsequent key findings. They are then interpreted in the light of existing theoretical and empirical knowledge. Novel findings should be highlighted. The interpretation of the results needs to be improved. The implications of the study results need to be highlighted.
Amended 4: The study aimed to investigate the effects of an MM programme, which combined elements of music and meditation, on two main outcomes: mindfulness and students' subjective mental happiness. The primary objectives were to understand how this programme impacted various aspects of participants' well-being, including their ability to cope with emotional difficulties and daily life stress, as well as their overall mental wellness. The key findings of the study revealed that the MM programme had a notably positive impact on the psychological well-being of the participants. Specifically, the programme led to significant improvements in participants' subjective happiness and mindfulness. Additionally, it was observed that the programme enhanced participants' capacity to handle emotional challenges and effectively manage stress in their daily lives. Overall, these findings underscored the programme’s effectiveness in promoting well-being, mindfulness, and mental wellness among the students involved in the study.
We believe that these revisions have significantly improved the quality and clarity of our manuscript. We appreciate your feedback, which has helped us enhance the overall rigour and comprehensibility of our research. Please let us know if you have any further suggestions or if there are any additional aspects that require attention.
Thank you for your time and consideration.
Sincerely,
Mi Hyang Hwang
Reviewer 2 Report (New Reviewer)
Comments and Suggestions for Authors
This is a thoughtfully planned study concerning the effects on short-term (during intervention) and longer-term (everyday life) experience of a 8-week zen meditation and music programme. The methods are qualitative, a semi-structured interview to assess mindfulness and happiness outcomes.
Here are a few comments:
Lines 147-152: Formatting
Lines 155ff: In the abstract it is mentioned that participants are music therapy students. It is important to write this down in the Methods section under participants.
Lines 178ff, Setting and procedure: Make explicit that there was a one-hour group session per week at the institution (where? at a university?). Participants were encouraged to do conduct the study also at home. What was the instruction (how many times, once a day, several times a week?).
Lines 186ff: It is mentioned that "some participants" (how many?) were unfamiliar with mindfulness meditation. Indicate the background of participants' meditation skills at the beginning of the study (see lines 693f: two participants used OM practice at home).
Line 196: typo in experience
Line 232: I would avoid the term "sedative" which is used for tranquilizers. The induced effect was to calm / relax.
In the Methods / Results: Make explicit that there is a distinction between "state" experience (during intervention) and "trait" changes (during everyday life). That is not clear at the beginning. This reviewer first thought that it was about changes in daily life but there are quotes of state experiences of the interventions. Maybe it is usufull to structure the Results and Discussion into these two domains: (1) how people experience the interventions and (2) how it affects daily life.
Lines 606ff: I would not worry about other studies that might have been conducted longer. The 8-week interval is the established standard.
Author Response
Dear Academic Editor,
Thank you for your valuable feedback on our manuscript. We have carefully considered your comments and have made the necessary revisions to address them. Here are our responses to each of your points:
Reviewer (2)
Q1: Lines 147-152: Formatting
Amended 1: We have amended
Q2: Lines 155ff: In the abstract it is mentioned that participants are music therapy students. It is important to write this down in the Methods section under participants.
Amended 2: The members of the group had diverse musical backgrounds and were familiar with therapy, as they were music therapy students.
Q3: Lines 178ff, Setting and procedure: Make explicit that there was a one-hour group session per week at the institution (where? at a university?). Participants were encouraged to do conduct the study also at home. What was the instruction (how many times, once a day, several times a week?).
Amended 3: (1) The session took place at the end of a teaching day at the university.
(2) Guided meditation files and music links, as used in the session, were sent via email for the students to practise at home once a day using their own streaming devices.
Q4: Lines 186ff: It is mentioned that "some participants" (how many?) were unfamiliar with mindfulness meditation. Indicate the background of participants' meditation skills at the beginning of the study (see lines 693f: two participants used OM practice at home).
Amended 4: Participants were interested in meditation practice; however, as there were various kinds of meditation used in this MM programme, more detailed guidance was provided. This included a short introduction to meditation and, when necessary, practical examples were demonstrated, such as 'Breathing Hand' and 'Breathing Clock.'
Q5: Line 196: typo in experience
Amended 5: sharing experience
Q6: Line 232: I would avoid the term "sedative" which is used for tranquilizers. The induced effect was to calm / relax.
Amended 6: Relaxing music
Q7: In the Methods / Results: Make explicit that there is a distinction between "state" experience (during intervention) and "trait" changes (during everyday life). That is not clear at the beginning. This reviewer first thought that it was about changes in daily life but there are quotes of state experiences of the interventions. Maybe it is usufull to structure the Results and Discussion into these two domains: (1) how people experience the interventions and (2) how it affects daily life.
Amended 7: During the 8-week programme, participants had various experiences of 'state' and 'trait' changes, and they discussed their experiences with the intervention and how it affected their daily lives.
Q 8: Lines 606ff: I would not worry about other studies that might have been conducted longer. The 8-week interval is the established standard.
I agree with you. The only issue is that during the sessions, I realized that some participants would have liked to have more time for deeper meditation. However, we had limited session time, so we needed to finish on time.
We believe that these revisions have significantly improved the quality and clarity of our manuscript. We appreciate your feedback, which has helped us enhance the overall rigour and comprehensibility of our research. Please let us know if you have any further suggestions or if there are any additional aspects that require attention.
Thank you for your time and consideration.
Sincerely,
Mi Hyang Hwang
This manuscript is a resubmission of an earlier submission. The following is a list of the peer review reports and author responses from that submission.
Round 1
Reviewer 1 Report
Comments and Suggestions for Authors
Thank you for the opportunity to review the article 'The Effect of Mindfulness and Music on Mental Wellness: A 2 Mixed Methods Study", by Mi hyang Hwang and the University of West of England music therapy department. Overall, I think this is a good study that can be published without major alterations.
My comments and critique are as follows.
1. The introduction refers at numerous place that researchers have discussed this or that aspect, relevant to the study. Yet they often don't say whether there were empirical findings in the study, or how methodologically robust they are. This seems to be to a minor degree a problem with the English of the article, which is not bad, but could do with a careful read-through and rephrasing or minor corrections e.g. by the senior author.
2. The hypotheses read like aims. They are not clearly operationalised - what does 'meaningful changes in the 6 factors of happiness' etc. mean? Does it mean that all 6 factors would change or the better? That the total score would? etc. I suggest condensing the hypotheses into a smaller number, focusing on specific ones intended by the authors.
3. The characteristics of the group are described, but key questions remain. Were these first-year music-therapy students who were all classical music trained to BMus level, for example? Did some have personal experience of therapy? Did they have knowledge of Mindfulness? Also, reporting these demographics in terms of percentages only in such a small sample is strange. How can 9.98 % of the sample have a particular characteristic - this is not an integral number of 9 people. I suggest reporting them as a fraction, e.g. 1/8 , 8/9 .
4. I found the mixed methods approach really good, giving different aspects understanding. Re. the quantitative results, the ones really hypothesized should be highlighted, and the exploratory ones either omitted, or described separately. For example, the question-by-question results of Table 5 are totally opaque (which reader knows which question says what?) and methodologically dubious.
5. With respect to the qualitative prompts, how were these determined? Some to be very nicely thought out, but not clearly reflected in the Results or Discussion (i.e., insight re. doing MM vs. doing M and M separately?) (NB the table contains erroneous numbering and the odd typo).
6. In the pdf copy available to me, Fig. 4 is strangely truncated and I think it is referred to as Fig 2 in the nearby text.
7. I found the quantitative results most intriguing. Neither the total nor the component scores of the Mindfulness measure showed pre-post change, but the total score of the Happiness measure did, as did several of its sub-scores (NB the tables contain typos). Yet this strange dissociation is not discussed. Of course, we could say that in such a small sample both false-positive and false-negative results would be common. However, if we stick with clinical thinking for a moment, if the specific Mindfulness content changed happiness, we would first observe a change in Mindfulness, and then, with greater difficulty, a change in Happiness. Putting it a little more statistically, Mindfulness change cannot statistically relate to Happiness change, if the former is not there.
This means that the authors need to discuss less of the detail of mindfulness and music, in the discussion, and more about the above.
My somewhat unkind guess is that the group therapy itself enabled participants to do more meaningfully and deeply what they already practiced.
8. Related to this, the absence of a control group needs to be discussed. Iiew,n fact, the study itself constitutes a pilot for thinking about what a control group would be. In my view, two interesting control groups could be included: a) Mindfulness programme, but app- and web-resources based, with minimal or no therapist input. This would approximate what students are nowadays very often encouraged to do. b) group, therapist and music listening only.
In the same vein, the possibility that the happiness improvements may reflect something completely unrelated (e.g., spending time in a lovely, supportive university doing a fantastically interesting course, and I'm not joking) need to be discussed.
The study should be compared and contrasted: https://mentalhealth.bmj.com/content/25/3/99.abstract , with particular attention to the fact that Ms. Huwang may be a vastly superior therapist to an app or a mindfulness-amateur Western therapist. Again this is no joke. In fact there are hints in the present discussion that non-specific factors, and especially the group setting, may have contributed to effectiveness.
9. Next, for this round of reviews, I would like to underline the importance of applicability and generalisability. I suspect that the particular model of therapy is culturally highly congruent with the group that was tested. As the authors know, successful therapy has been thought to have 'A rationale, or myth, which includes an explanation of the cause of the patient’s distress and a method for relieving it. To be effective, the therapeutic myth must be compatible with the cultural
world-view shared by patient and therapist. ' (Originally Frank, copied from https://focus.psychiatryonline.org/doi/pdf/10.1176/foc.4.2.306 ) In my humble experience, sophisticated mindfulness practices may fall on deaf ears among specific (including cynical!) social groups, and so can outgroup music.
I bet that by now the UWE is doing studies, or even giving treatments, based on online-group versions of MM? Why not mention this as future research?
Finally a point which is most probably impractical, but is worth a shot. It may be worth re-contacting an briefly interviewing the group as to the long-term effect the therapy had on mindfully experiencing music during the pandemic, and even comparing this with students who have been introduced to mindfulness at around the same period but did not have the enriching experience of MM.
Overall, I would like to congratulate Hwang, Bunt and Warner on their contribution and hope to see it published soon. My recommendation falls between 'accept with minor revision' and 'Reconsider after major revision' in the choices I was given, but given that the latter specifically mentions control group (which I do not think is sine qua non here, but it is very important to take seriously, not just rely on before-after), I recommend major revision.
Reviewer 2 Report
Comments and Suggestions for Authors
The detailed manuscript review report is attached below.

Reviewer 3 Report
Comments and Suggestions for Authors
In this manuscript, a mixed-method study investigating the effect of mindfulness and music therapy is assessed through both questionnaires and semi-structured interviews. The results seem to support that this intervention is effective to increase happiness. However, this study has a serious and unnegotiable flaw. When I read the participants' section, I understand that only nine participants took part in this intervention. Given this number, the study has a potential power of about 0.25 (from a rapid G-power analysis). Thus, it could not be accepted nor considered a piece of scientific literature.
My suggestion is to enlarge your sample based on proper power analysis, and report effect size along with the significance of results.
I am sorry, but given this fatal flaw, this paper could not be considered further. That is why I would not report here a more detailed review.
Round 2
Reviewer 3 Report
Comments and Suggestions for Authors
Dear Authors,
the minor amendments to the text and especially the introduction are not sufficient to let me consider this manuscript as a report of scientific results. Again, the complete lack of power analysis and effect size report completely undermines every possible value of this manuscript. I am sorry to seem so rude, as I also appreciate your intervention, but this cannot be considered in any case a scientific contribution.